# RSA: Reducing Semantic Shift from Aggressive Augmentations for Self-supervised Learning

**Yingbin Bai**[1]    **Erkun Yang**[2]    **Zhaoqing Wang**[1]
**Yuxuan Du**[3]    **Bo Han**[4]    **Cheng Deng**[2]    **Dadong Wang**[5]    **Tongliang Liu**[1] *

[1]TML Lab, The University of Sydney; [2]Xidian University; [3]JD Explore Academy;
[4]Hong Kong Baptist University; [5]CSIRO

## Abstract

Most recent self-supervised learning methods learn visual representation by contrasting different augmented views of images. Compared with supervised learning, more aggressive augmentations have been introduced to further improve the diversity of training pairs. However, aggressive augmentations may distort images' structures leading to a severe semantic shift problem that augmented views of the same image may not share the same semantics, thus degrading the transfer performance. To address this problem, we propose a new SSL paradigm, which counteracts the impact of semantic shift by balancing the role of weak and aggressively augmented pairs. Specifically, semantically inconsistent pairs are of minority, and we treat them as noisy pairs. Note that deep neural networks (DNNs) have a crucial memorization effect that DNNs tend to first memorize clean (majority) examples before overfitting to noisy (minority) examples. Therefore, we set a relatively large weight for aggressively augmented data pairs at the early learning stage. With the training going on, the model begins to overfit noisy pairs. Accordingly, we gradually reduce the weights of aggressively augmented pairs. In doing so, our method can better embrace aggressive augmentations and neutralize the semantic shift problem. Experiments show that our model achieves 73.1% top-1 accuracy on ImageNet-1K with ResNet-50 for 200 epochs, which is a 2.5% improvement over BYOL. Moreover, experiments also demonstrate that the learned representations can transfer well for various downstream tasks. Code is released at: `https://github.com/tmllab/RSA`.

## 1   Introduction

A golden law in the context of computer vision is utilizing tremendous annotated data to learn good visual representations [58, 26]. Unfortunately, collecting annotated data with accurate labels is generally laborious, expensive [52, 56], and even infeasible [22]. To this end, various approaches have been proposed to learn such representations from unlabelled visual data, usually by performing visual pretext tasks. Among them, self-supervised learning methods [6, 7, 46, 9] based on contrastive loss have recently shown great promise, achieving state-of-the-art performance.

Representative contrastive methods are generally trained by maximizing agreement between differently augmented views of the same image (positive pairs), and increasing the distance between augmented views from different images (negative pairs) [49, 6, 17]. Compared with supervised learning, these works highlight the role of data augmentation for SSL and design more aggressive augmentation operations. Here, we refer aggressive augmentations as the operations that can possibly change the semantics of images, such as grayscale, color jitter, and Gaussian blur. Other used

---

*Correspondence to Tongliang Liu (tongliang.liu@sydney.edu.au)

36th Conference on Neural Information Processing Systems (NeurIPS 2022).

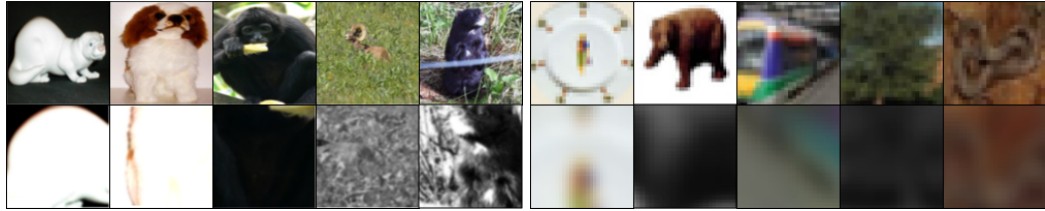

| (a) Noisy samples from ImageNet-1K [25] | (b) Noisy samples from CIFAR-100 [24] |

Figure 1: The first row is the original images (ImageNet-1K images are resized to squares) and noisy samples from aggressive augmentation are in the second row. From the first three images of (a), we can observe that Color jitter operation makes the image too bright or too dark that covers the details of images; and in the fourth and fifth columns of (a), Grayscale and Gaussian blur operation leads images to be hardly distinguished from the background; and the same augmentation strategy from ImageNet-1K leads to more vague images for CIFAR-100 shown in (b).

augmentations are referred as weak augmentations. It should be noted that the random cropping operation can also change the semantic information, however, this issue can be properly addressed by exploiting object detection techniques [40, 35]. Therefore, we do not include the random cropping operation as an aggressive augmentation. Although the aggressive augmentations can help to further improve the model performance, they also bring a severe semantic shift problem for training images. As illustrated in Figure 1, the first row shows original images from ImageNet [25] and CIFAR-100 [24] datasets. And the second row presents the corresponding augmented views with the widely used composition of augmentations [7, 6]. We can see that the augmented views can be hardly recognized as semantically consistent with their original versions. Pushing these images to have similar representations can adversely affect the model training, so we consider the semantically inconsistent pairs as noisy pairs. However, due to the diversity of training images and the randomness of augmentation, it is difficult to accurately measure the quality of augmented pairs. Attempting to roughly remove noisy pairs may result in a decrease in performance since they are mixed with clean (semantic consistent) pairs.

Fortunately, recent works [2, 57, 16, 21, 30] show that deep neural networks (DNNs) have a crucial memorization effect that DNNs tend to first memorize clean (majority/semantically consistent) examples before overfitting noisy (minority/semantically inconsistent) examples. Motivated by this, we propose a method called Reducing Semantic shift from Aggressive augmentation (RSA) by dynamically adjusting the weights of clean and noisy pairs. Specifically, we introduce aggressive-weak augmented pairs as relatively clean pairs because weak augmentation has a low probability of causing the semantic shift problem, and set a relatively large weight for the aggressively augmented data pairs at the beginning of training to fully exploit all the training examples. As the training goes on, the model begins to overfit semantically inconsistent data. Therefore we gradually decrease the weight of aggressive augmented pairs and increase the weight of aggressive-weak augmented pairs to reduce semantically inconsistent impact.

Empirical results on multiple benchmark datasets show that our method can outperform state-of-the-art methods in various settings with a large margin. For instance, with 200 epochs of pre-training, our method achieves 73.1% Top-1 accuracy on ImagetNet-1K [25] linear evaluation protocol, which is 2.5% higher than BYOL [15]. Experiments on MS COCO [29] also show that our pre-trained model can continually improve the performance for multiple downstream tasks.

## 2   Related Works

**Self-supervised learning (SSL)** has attracted great attention to capture universal representations [20, 60, 38, 48, 5, 12, 61]. The core of SSL is designing agent tasks, which allow us to learn representations from large-scale unlabeled data via pseudo labels instead of using any human annotations. To this end, many proposals devise different solutions for constructing pseudo labels, including predicting the rotation of images [13], putting pieces of images together [36], or recovering color from grayscale images [59]. Particularly, Wu et al. [49] propose an instance-level classification, which regards images augmented from the same image as a positive pair and others as negative examples. SimCLR [6]

improves performance by inserting the projection network and introducing aggressive augmentation. He et al. [17, 7] store negative representations in a queue to reduce the memory requirement. BYOL [15] enhances the power of SSL by removing the dependence on negative examples, which also addresses the problem of false negative examples [9]. Despite these methods having proved their effectiveness based on aggressive augmentation, they ignore the semantic shift problem from it.

**Noisy samples in aggressive augmentation**. Recent studies [47, 40, 35, 39] have discovered that aggressive augmentation may generate noisy samples in the positive pairs. To alleviate this issue, ContraCAM [35] proposes a two-step approach to reduce the issue of random cropping, which seeks objects first and then crops images based on their locations. In addition, Gansbeke et al. [45] conduct experiments on scene-centric datasets (e.g., COCO) containing multiple objects in images and argue that SSL can overcome the issue of random cropping. Since the issue of random cropping has been well studied, our work focuses on other types of augmentations, which can be viewed as a complement to previous studies.

**Learning with noisy labels**. Reducing the semantic shift problem in SSL is similar to another well-studied topic in machine learning, learning with noisy labels [31, 51, 54, 50, 53]. In this field, many state-of-the-art methods use the memorization effect to select confident examples in the early learning phase [37, 41, 27]. Specifically, Co-teaching [16] uses the early stopping trick and the small-loss strategy to choose confident examples. PES [3] finds examples with noisy labels have more detrimental effects on the latter layers, and improves the early stopping trick by progressively training each part of DNNs. However, directly using the early stopping trick is difficult in SSL because noisy pairs are randomly generated by aggressive augmentations over epochs while samples with noisy labels are fixed in learning with noisy labels. Besides, roughly removing noisy pairs is more likely to delete many clean pairs by mistake, especially in the low noise rate case.

## 3 Methodology

Our approach aims to minimize the detrimental impacts of semantically inconsistent (noisy) pairs from aggressive augmentations while taking advantage of aggressive augmentations. As such, we first revisit preliminaries on self-supervised learning. Then, we elaborate on the proposed learning algorithm that counterbalances the noise impacts by utilizing the memorization effect of DNNs.

### 3.1 Preliminaries on Self-supervised Learning

Self-supervised learning methods based on contrastive learning generally require learning an embedding space that can easily separate different examples. Let $D$ be a set of images, and an image $x_i$ is uniformly drawn from $D$. Denote $t$ and $t'$ as two different instances from the same distribution of image augmentation T. $v$ and $v'$ are two augmented views of the image $x_i$ with $v = t(x_i)$ and $v' = t'(x_i)$, which are regarded as a pair of positive examples. Then, $v$ and $v'$ will separately feed through an encoder $f_\theta$ and a projector $g_\theta$ to embed $z$ and $z'$, which are required to close to each other via a contrastive loss function, e.g., InfoNCE [44] can be expressed as,

$$\mathcal{L}_{NCE} = -\log \frac{\exp(z \cdot z'/\gamma)}{\exp(z \cdot z'/\gamma) + \sum_{n \in N} \exp(z \cdot n/\gamma)}, \tag{1}$$

where $\gamma$ is a temperature parameter, and N is a set of negative example vectors. The embeddings of positive and negative examples are $l_2$-normalized. To stabilize the training process, some state-of-the-art methods [17, 61, 23] employ an asymmetric framework, including an online network and a target network. For the online and target networks, they have the same network structure but different weights of encoder $f_\xi$, and projector $g_\xi$, whose parameters $\xi$ are updated by the online parameters $\theta$ with the exponential moving average method.

Recently, BYOL [15] found negative examples are not necessary and added a predictor $q_\theta$ in the online network to avoid collapsed solutions, e.g., all images have the same vector. And, the loss function can be simplified to,

$$\mathcal{L}_{mse} = 2 - 2 * \frac{\langle z, z' \rangle}{\|z\|_2 * \|z'\|_2}. \tag{2}$$

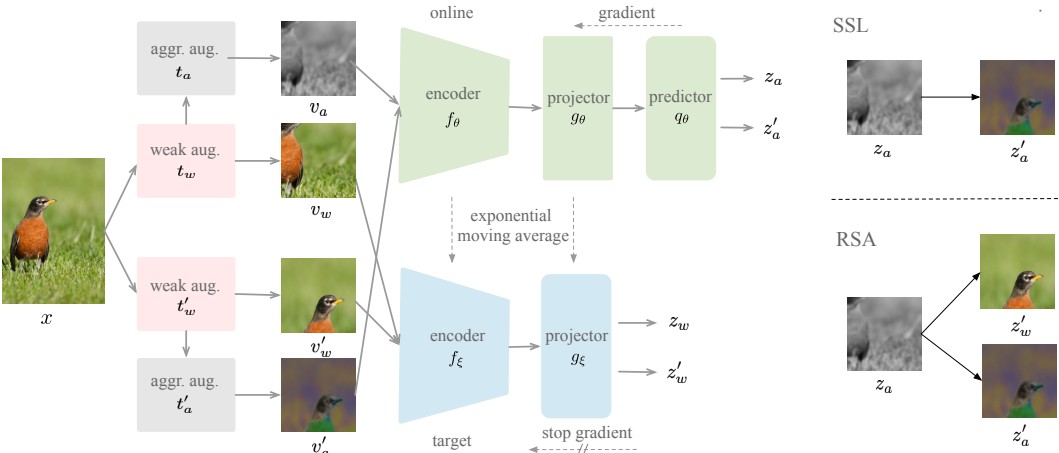

Figure 2: The illustration of our proposed method (RSA). We utilize an asymmetric-style framework, including an online network and a target network. The online network is optimized by gradients, and the target network is updated with the exponential moving average strategy. We first adopt the weak augmentation to generate two views $(v_w, v'_w)$, then adopt the aggressive augmentations to further generate another two views $(v_a, v'_a)$. Subsequently, we make aggressive-augmented views to keep consistent with their corresponding weak- and aggressive-augmented views in the embedding space. On the right of the image, we compare RSA with classical SSL methods. RSA forces learned representations to a balance between weak- and aggressive-augmented views. (Best viewed in color)

## 3.2 Description of RSA

As discussed in Introduction, noisy pairs from aggressive augmentations will lead to a severe semantic shift problem, which damages the generalization on downstream tasks. However, most of the sample pairs generated from aggressive augmentations are beneficial for the model performance, and what precise degree of distortion would cause bias remains unknown, so it is hard to distinguish between clean pairs and noisy pairs in SSL. Instead of separating noise from training data, we employ weak augmentation to generate sample pairs as relatively clean pairs.

To this end, we propose an efficient method to reduce semantic shift from aggressive augmentations, dubbed RSA. Envisioned by the memorization effect, DNNs will first fit clean pairs (majority/semantically consistent) of training data in the early learning phase and then overfit to noisy pairs (minority/semantically inconsistent). Noisy pairs account for the minority, so we assume that the noise impacts will be small and then increases as the training process. We give different weights to clean and noisy pairs, and gradually reduce the weights of noisy pairs against raised noise impacts.

However, simply introducing clean pairs will significantly increase computation, and SSL has known that more computation will increases performance, which makes it difficult to fairly compare with other baselines. To limit computation, we meet two problems. First, generating more augmented instances, especially double instances, commonly means expensive computation, largely burdening CPU and hard disk, which leads to slow data processing down and low efficiency of GPU [6]. Second, more instances require more back-propagation times, which results in more processing time for GPU.

To address the first problem, we propose a novel data augmentation pipeline, called multi-stage augmentation, which can generate double instances with nearly the same resources. Specifically, data augmentation generates image variants by mixing different types of image transformation and arranging them into a queue. For a transformation, it receives an image from the result of the previous transformation and transforms the image based on pre-defined probability, and then passes it to the next transformation, $t_{aug}(x) = t_a(t_w(x))$, where $t_a$ and $t_w$ are two subsets of $t_{aug}$. Based on this process, we can separate it into two child processes. Namely, $t_w$ includes weak augmentation operations while $t_a$ conducts aggressive augmentation operations based on $t_w$. This process can be described as,

$$v_w = t_w(x) \quad v'_w = t'_w(x),$$
$$v_a = t_a(v_w) \quad v'_a = t'_a(v'_w). \tag{3}$$

**Algorithm 1:** RSA: Reducing semantic shift

---

**Input**: Neural network $f_\theta$ and $f_\xi$. Projector $g_\theta$ and $g_\xi$. Predictor $q_\theta$. A batch of samples $x$. An image set $D$. Total number of training steps $K$. Weak augmentation function $t_w$. Aggressive augmentation function $t_a$. Hyper-parameter $\beta_{base}$.

**for** $k \leftarrow 1$ **to** $K$ **do**

    $x \leftarrow D$                                             `// Sample a batch of images`

    $v_w \leftarrow t_w(x)$ and $v'_w \leftarrow t'_w(x)$                       `// Multi-stage augmentation`

    $v_a \leftarrow t_a(v_w)$ and $v'_a \leftarrow t'_a(v'_w)$

    $z_a \leftarrow q_\theta(g_\theta(f_\theta(v_a)))$ and $z_w \leftarrow g_\xi(f_\xi(v_w))$        `// Embedding for four views`

    $z'_a \leftarrow q_\theta(g_\theta(f_\theta(v'_a)))$ and $z'_w \leftarrow g_\xi(f_\xi(v'_w))$

    $\mathcal{L} \leftarrow (1-\beta)\mathcal{L}_{mse}(z_a, z'_w) + \beta\mathcal{L}_{mse}(z_a, z'_a) + (1-\beta)\mathcal{L}_{mse}(z'_a, z_w) + \beta\mathcal{L}_{mse}(z'_a, z_a)$

    $\beta \leftarrow \beta_{base} \times \frac{1}{2}(cos(\pi\frac{k}{K}) + 1)$                        `// Decreasing` $\beta$

    Update $f_\theta$, $g_\theta$ and $q_\theta$ with $\mathcal{L}$

    Update $f_\xi$ and $g_\xi$ by slowly momentum updating with $f_\theta$ and $g_\theta$

**Output**: The trained network $f_\theta$

---

For the second problem, we send aggressive augmented views to the online network and weak augmented views to the target network, which means each instance only forwards once. Accordingly, RSA requires back-propagation twice, which is equal to many state-of-the-art SSL methods with a symmetrized loss [6, 8, 15, 5].

$$\begin{aligned} z_a = q_\theta(g_\theta(f_\theta(v_a))) \quad &z_w = g_\xi(f_\xi(v_w)), \\ z'_a = q_\theta(g_\theta(f_\theta(v'_a))) \quad &z'_w = g_\xi(f_\xi(v'_w)). \end{aligned} \tag{4}$$

After obtaining four representations, we group them into four pairs and make each aggressive-augmented view to keep consistent with their corresponding weak- and aggressive-augmented views in the embedding space. For the corresponding aggressive-augmented views, rather than encoding them on the target network, we use the representations from the online network with a stop-gradient operation. The total loss is summarized as,

$$\begin{aligned} \mathcal{L} = (1-\beta)\mathcal{L}_{mse}(z_a, z'_w) + \beta\mathcal{L}_{mse}(z_a, z'_a) \\ + (1-\beta)\mathcal{L}_{mse}(z'_a, z_w) + \beta\mathcal{L}_{mse}(z'_a, z_a), \end{aligned} \tag{5}$$

where $\beta$ is a calculated parameter to re-weight the losses from weak- and aggressive-augmented views. This loss function forces networks to learn representations that achieve a balance between weak- and aggressive-augmented views, which reduces overfitting to noisy pairs while avoiding a simple solution. As the training goes on, the noisy impacts will increase. We further offset it by using the memorization effect, which decreases $\beta$ with a cosine decay equation,

$$\beta = \beta_{base} \times \frac{1}{2}(cos(\pi\frac{k}{K}) + 1), \tag{6}$$

where $\beta_{base}$ is a given number at the training beginning, and k and K are the current training steps and the total training steps.

## 4 Experiments

### 4.1 Datasets and Implementation Details

**Datasets:** We assess the proposed method on six image datasets, from small to large. We choose CIFAR-10/100 [24] for small datasets, and STL-10 [10] and Tiny ImageNet [1] for medium datasets, and ImageNet-100 [42] and ImageNet-1K [25] for large datasets. Note, ImageNet-100 contains 100

Table 1: Analysis of the multi-stage augmentation and the memorization effect. We run methods on small and medium datasets for 200 epochs, and on ImageNet-100 for 100 epochs. The mean and standard deviation are computed over three trials. Note that two augmented instances with aggressive augmentation are referred to as AA, instances with one aggressive and one weak augmentation as AW, and instances with multi-stage augmentation as MA.

| Method | Aug. | $\beta$ | CIFAR-10 | CIFAR-100 | STL-10 | Tiny ImageNet | ImageNet-100 |
|--------|------|---------|----------|-----------|--------|---------------|--------------|
| BYOL | AA | – | 90.2±0.0 | 63.9±0.5 | 91.3±0.1 | 49.5±0.1 | 80.9 |
| BYOL | AW | – | 90.5±0.2 | 66.1±0.3 | 90.7±0.0 | 51.1±0.3 | 81.2 |
| RSA | MA | Fixed | 91.3±0.2 | 66.5±0.0 | 92.8±0.2 | 53.9±0.2 | 83.2 |
| RSA | MA | Decay | **92.1±0.0** | **67.6±0.3** | **93.0±0.2** | **54.7±0.3** | **83.5** |

classes that are randomly selected from ImageNet-1K [25] and we choose the same classes with [42]. For STL-10, both 5k labeled and 100k unlabeled images are used for the pre-trained model, and only 5k labeled images are used for the linear evaluation.

**Augmentation:** In this paper, we define "aggressive" augmentation including grayscale, color jitter, and Gaussian blur, while "weak" augmentation includes random crop and horizontal flip. The hyper-parameters of the augmentations are following MoCo v2 [7] except for the size of the cropped images for the small and medium datasets, and we resize images to $32 \times 32$ and $64 \times 64$ for the small and medium datasets, respectively.

**Baselines:** For the comparison, we re-implement the state-of-the-art methods, SimCLR [6], MoCo v2 [7], SimSiam [8], and BYOL [15] based on the public codes. We follow [8] that implements the MoCo v2 with a symmetrized loss function, and set the exponential moving average factor to 0.99 for all experiments. For BYOL [15] on the small and medium datasets, we follow [15], and set the channel inner layer of 1024 in the projection and prediction MLP and the output feature is 128. We initially set the exponential moving average factor as 0.99 and gradually enlarge it to 1.

**Network structure and optimization:** Our method and reproduced methods are implemented by PyTorch v1.8 and we conduct all experiments on Nvidia V100. Our method is based on our reproduced BYOL [15]. For the pre-train stage on the small and medium datasets, we adopt ResNet-18 [19] as a backbone. For optimization, we use SGD optimizer with a cosine-annealed learning rate of 0.1 [32], a momentum of 0.9, weight decay of $5 \times 10^{-4}$, and a batch size of 256. We set $\beta_{base} = 0.3$ for CIFAR-10/100 and $\beta_{base} = 0.4$ for Tiny ImageNet and STL-10.

For the pre-train stage on large datasets, we adopt a standard ResNet-50 [19] as a backbone. For ImageNet-100, the network is trained using SGD optimizer with a single cycle of cosine annealing [32], and an initial learning rate of 0.2, a momentum of 0.9, weight decay of $10^{-4}$, and a batch size of 256. We conduct ImageNet-1K experiments with $8 \times$ Nvidia V100 32G with Automatic Mixed Precision (AMP) package [33]. Specifically, we follow [15], and train a network with a LARS optimizer [55] with a single cycle of cosine annealing [32], a momentum of 0.9, weight decay of $10^{-6}$, and a batch size of 2048. The base learning rate starts from 0.9 and 0.6 for 100 and 200 epochs respectively, linearly scaled by the times of batch size 256 [14]. We set $\beta_{base} = 0.4$ for ImageNet-100 and ImageNet-1K.

**Evaluation:** We evaluate the representations of the pre-trained model with the linear evaluation protocol, which freezes the encoder parameters and trains a linear classifier on top of the pre-trained model. For the small and medium datasets, we follow the setting in MoCo v2 [7] and train a linear classifier for 100 epochs with an initial learning rate of 30, no weight decay, and a momentum of 0.9. The learning rate will be multiplied by 0.1 at the 60 and 80 epochs. For the large datasets, we follow the evaluation setting in Mean Shift [23], which only requires 40 epochs and a batch size of 256. The linear classifier is trained with SGD and an initial learning rate of 0.01, weight decay of $10^{-4}$, and a momentum of 0.9. The learning rate will be multiplied by 0.1 at 15, 30, and 40 epochs.

### 4.2 Preliminary Analysis

In Table 1, we investigate the effectiveness of the proposed multi-stage augmentation and the memorization effect. We first compare the linear classification of BYOL with two aggressive augmentations (AA) and with one aggressive and one weak augmentation (AW) and RSA with multi-stage augmentation (MA) without memorization effect (fixed $\beta$). From the second row, we can

Table 2: Performance comparison with linear classification on small and medium datasets for 200 and 800 epochs. We adopt a ResNet-18 as a backbone for all experiments. The mean and standard deviation are computed over three trials.

| Method | CIFAR-10 | | CIFAR-100 | | STL-10 | | Tiny ImageNet | |
|---|---|---|---|---|---|---|---|---|
| | 200 ep | 800 ep | 200 ep | 800 ep | 200 ep | 800 ep | 200 ep | 800 ep |
| SimCLR | 88.3±0.2 | 91.1 | 59.5±0.3 | 63.8 | 87.9±0.4 | 91.0 | 46.7±0.1 | 48.4 |
| MoCo v2 | 87.1±0.2 | 91.4 | 60.3±0.4 | 65.0 | 88.4±0.3 | 91.3 | 47.8±0.4 | 50.1 |
| SimSiam | 86.9±0.1 | 91.1 | 55.8±0.9 | 62.4 | 85.0±0.4 | 89.7 | 41.2±0.3 | 44.5 |
| BYOL | 90.2±0.0 | 92.7 | 63.9±0.5 | 68.2 | 91.3±0.1 | 93.4 | 49.5±0.1 | 53.0 |
| RSA (Ours) | **92.3±0.1** | **93.7** | **68.1±0.5** | **70.4** | **93.1±0.2** | **94.0** | **54.8±0.4** | **55.5** |

see that BYOL with AW can largely improve the performance on CIFAR-100 and Tiny ImageNet, and mildly improve the performance on CIFAR-10 and ImageNet-100, but the network suffers from AW on STL-10. These inconsistent results suggest that although AW can help against the noise impacts from aggressive pairs, it reduces the diversity of augmented pairs compared with AA, which may result in a suboptimal result. In contrast, MA not only outperforms AA and AW but also generally improves performance across five datasets, suggesting that multi-stage augmentation (MA) can take advantage of AA and AW at the same time.

According to the memorization effect of DNNs, noise impacts vary at different stages of the training process and become more apparent at the end. In order to balance the increasing noise impact, we decay $\beta$ during the training, which results in rising the weights of aggressive-weak augmented pairs and reducing the weights of aggressive-aggressive augmented pairs. Improved performance in the fourth row verifies that we can use the memorization effect of DNNs to further reduce noise impact. Notably, we set $\beta_{base} = 0.5$ in preliminary experiments, and the performance can be further improved with a turned $\beta_{base}$ in Table 2 and 3.

## 4.3 Linear Classification

**Small and medium datasets.** We evaluate our method on small and medium datasets, with 200 and 800 epochs. We also run three trials for a short running time to evaluate the stability of the proposed method. Table 2 illustrates that the proposed method significantly improves the performance across the four datasets on short training time experiments, demonstrating that RSA can accelerate convergence and the small standard deviation also shows that the proposed method has good stability. For the long training time experiments, outstanding results show RSA can continue to improve the final performance.

Table 3: Performance comparison with linear classification on ImageNet-100 for 100 and 200 epochs. All methods adopt ResNet-50 as a backbone.

| Method | Batch size | 100 ep | 200 ep |
|---|---|---|---|
| SimCLR | 256 | 79.1 | 82.4 |
| MoCo v2 | 256 | 80.9 | 83.9 |
| SimSiam | 256 | 79.7 | 82.6 |
| BYOL | 256 | 80.9 | 83.6 |
| RSA (Ours) | 256 | **83.7** | **85.5** |

**Large datasets.** We evaluate the performance of the proposed method on the large datasets, ImageNet-100 and ImageNet-1K. We reproduce all baselines on ImageNet-100 with batch size 256. The results on ImageNet-100 and ImageNet-1K are shown in Table 3 and Table 4 respectively. We can see that RSA outperforms the state-of-the-art methods on ImageNet-100 with a relatively large margin across 100 and 200 epochs. For results on ImageNet-1K, RSA consistently surpasses baselines, e.g., the performance of RSA for 100 epochs has already surpassed BYOL training for 200 epochs. RSA achieves a new state-of-the-art result for 200 epochs, exhibiting a 2.5% improvement over BYOL. Overall, empirical results on linear evaluation verify that RSA can constantly improve the generalization in various settings.

## 4.4 Transfer Learning

We further verify the quality of representation learned by RSA on more downstream tasks. For object detection and instance segmentation, we follow [43], and adopt Mask R-CNN [18] with FPN [28] to fine-tune our pre-trained ResNet-50 model on COCO *train2017* with 1 × schedule and 2 × schedule,

Table 4: Performance comparison with linear classification on ImageNet-1K. All methods use a standard ResNet-50 as a backbone without a multi-crop strategy.

| Method | Neg. pairs | Batch Size | Epochs | Top-1 Linear |
|---|---|---|---|---|
| Supervised | | 256 | 100 | 76.2 |
| InstDis [49] | ✓ | 256 | 200 | 56.5 |
| PIRL [34] | ✓ | 256 | 200 | 63.6 |
| SimCLR [6] | ✓ | 4096 | 1000 | 69.3 |
| MoCo v2 [7] | ✓ | 256 | 200 | 67.5 |
| JCL [4] | ✓ | 256 | 200 | 68.7 |
| ReSSL [61] | ✓ | 256 | 200 | 69.6 |
| InfoMin Aug. [43] | ✓ | 256 | 200 | 70.1 |
| W-MSE 4 [12] | | 4096 | 400 | 72.6 |
| SimSiam [8] | | 256 | 200 | 70.0 |
| SwAV [8] | | 4096 | 200 | 69.1 |
| BYOL [8] | | 4096 | 100 | 66.5 |
| BYOL [8] | | 4096 | 200 | 70.6 |
| BYOL [15] | | 2048 | 300 | 72.4 |
| RSA (Ours) | | 2048 | 100 | 71.4 |
| RSA (Ours) | | 2048 | 200 | **73.1** |

Table 5: **Transfer learning on downstream tasks:** object detection, instance segmentation. All models were pre-trained on ImageNet-1K for 200 epochs and fine-tuned on MS COCO with 1 × schedule. Object detection and instance segmentation results are from [43].

| Method | Object detection | | | Instance segmentation | | |
|---|---|---|---|---|---|---|
| | $AP^{bb}$ | $AP^{bb}_{50}$ | $AP^{bb}_{75}$ | $AP^{mk}$ | $AP^{mk}_{50}$ | $AP^{mk}_{75}$ |
| random init. | 32.8 | 50.9 | 35.3 | 29.9 | 47.9 | 32.0 |
| supervised | 39.7 | 59.5 | 43.3 | 35.9 | 56.6 | 38.6 |
| InstDis [49] | 38.8 | 58.4 | 42.5 | 35.2 | 55.8 | 37.8 |
| PIRL [34] | 38.6 | 58.2 | 42.1 | 35.1 | 55.5 | 37.7 |
| MoCo [17] | 39.4 | 59.1 | 42.9 | 35.6 | 56.2 | 38.0 |
| MoCo V2 [7] | 40.1 | 59.8 | 44.1 | 36.3 | 56.9 | 39.1 |
| InfoMin Aug. [43] | 40.6 | 60.6 | 44.6 | 36.7 | 57.7 | 39.4 |
| RSA (Ours) | **41.1** | **61.4** | **45.1** | **37.3** | **58.6** | **40.1** |

Table 6: **Transfer learning on downstream tasks:** object detection, and instance segmentation. All models were pre-trained on ImageNet-1K for 200 epochs and fine-tuned on MS COCO with 2 × schedule. Baseline results are from [43].

| Method | Object detection | | | Instance segmentation | | |
|---|---|---|---|---|---|---|
| | $AP^{bb}$ | $AP^{bb}_{50}$ | $AP^{bb}_{75}$ | $AP^{mk}$ | $AP^{mk}_{50}$ | $AP^{mk}_{75}$ |
| Random | 38.4 | 57.5 | 42.0 | 34.7 | 54.8 | 37.2 |
| Supervised | 41.6 | 61.7 | 45.3 | 37.6 | 58.7 | 40.4 |
| InstDis [49] | 41.3 | 61.0 | 45.3 | 37.3 | 58.3 | 39.9 |
| PIRL [34] | 41.2 | 61.2 | 45.2 | 37.4 | 58.5 | 40.3 |
| MoCo [17] | 41.7 | 61.4 | 45.7 | 37.5 | 58.6 | 40.5 |
| MoCo v2 [7] | 41.7 | 61.6 | 45.6 | 37.6 | 58.7 | 40.5 |
| InfoMin Aug. [43] | 42.5 | 62.7 | 46.8 | 38.4 | 59.7 | **41.4** |
| RSA (Ours) | **42.7** | **62.9** | **47.1** | **38.5** | **60.0** | **41.4** |

and evaluate performance on COCO *val2017*. We set the initial learning rate as 0.02 and 0.03 for the 1 × schedule and 2 × schedule experiments respectively.

Table 5 and Table 6 report the results of object detection and instance segmentation. We can observe that RSA outperforms all baselines on object detection and instance segmentation tasks, especially, showing superior over the strong baseline InfoMin Aug. [43] that adopts more advanced augmentation, RandAugment [11]. Strong performance on downstream tasks demonstrates that RSA can improve the general quality of learned representations.

Table 7: Training time comparison with BYOL on three datasets

| Method | CIFAR-100 | STL-10 | ImageNet-100 |
|--------|-----------|--------|--------------|
| BYOL   | 11.3h     | 77.6h  | 9.5h         |
| RSA    | 11.5h     | 79.6h  | 10.4h        |

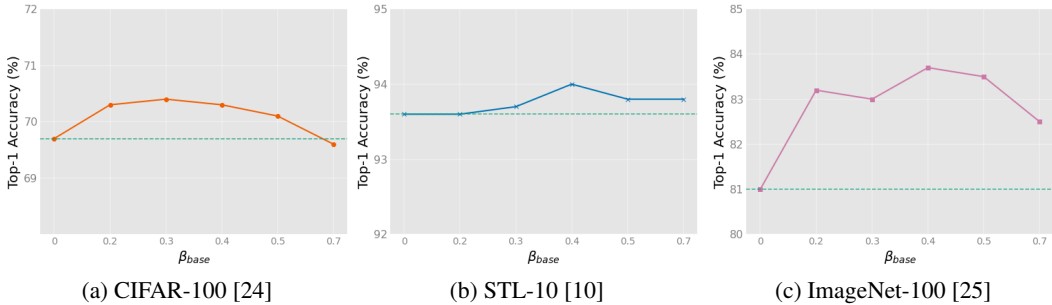

(a) CIFAR-100 [24]          (b) STL-10 [10]          (c) ImageNet-100 [25]

Figure 3: Sensitivity analysis for the hyper-parameter $\beta_{base}$. We conduct experiments on CIFAR-100 and STL-10 for 800 epochs, and ImageNet-100 for 100 epochs, respectively. Note that the green dash line is the result with $\beta = 0$.

## 4.5 Training Time Comparison

We compare the running time between our method and reproduced BYOL. We conduct experiments on CIFAR-100 and STL-10 for 800 epochs with a single Nvidia V100, and ImageNet-100 for 200 epochs with $4\times$ Nvidia V100, respectively. For ImageNet-100, we use Automatic Mixed Precision package [33] to speed up the training process and save GPU memory. Note that since we do not have enough GPUs to run a standard BYOL with a batch size of 4096 on ImageNet, we use ImageNet-100 instead of ImageNet-1K, which has similar processing efficiency but ten times the data.

Although our method uses double instances in the loss function 5, each instance in RSA only passes into the network one time. Therefore, the number of forward and backward passes keeps the same with BYOL. In addition, thanks to the efficient multi-stage augmentation, where we generate double instances using the nearly same resources. The main burden training time part of our proposed method may come from two more vector multiplication in our loss function 5. From Table 7, we can see that our method is as efficient as BYOL, and the differences between the two methods are less than 9% across three datasets.

## 4.6 Ablation Studies

In this section, we investigate the importance of the hyper-parameter $\beta_{base}$ in Eq 6, which controls the initial weights of aggressive-aggressive and aggressive-weak augmented pairs in Eq 5. Figure 3 reports the results with linear classification on three datasets. First of all, we can observe that $\beta_{base}$ makes remarkable contribution for the improvements compared with $\beta_{base} = 0$, with +0.4% (total increments +0.6%) for STL-10 and +0.7% (total increments +2.2%) for CIFAR-100. This suggests that removing aggressive-aggressive augmented pairs with $\beta_{base} = 0$ will limit the full exploration of the diversity of aggressive augmentation. However, if $\beta_{base}$ is too large, aggressive-weak augmented pairs will make less contribution against noisy impacts.

## 4.7 Adaptability Studies

To evaluate the adaptability, we implement the proposed method based on SimSiam [8], termed SimSiam + RSA. Specifically, we change the same similarity function from mean square error to negative cosine similarity and adopt the same settings for all the hyper-parameters mentioned in Section 4.1. As illustrated in Table 8, we observe that the proposed Simsiam + RSA achieves better transfer performance than SimSiam on three datasets. For instance, RSA significantly raises the accuracy of linear probing from 55.8% to 63.7% (+7.9%) on the CIFAR-100 dataset. Therefore, the proposed method does not rely on the momentum network, demonstrating the adaptability of RSA.

Table 8: Performance comparison with linear classification for 200 epochs. The mean and standard deviation are computed over three trials.

| Method | CIFAR-100 | STL-10 | ImageNet-100 |
|---|---|---|---|
| SimSiam | 55.8±0.9 | 85.0±0.4 | 82.6 |
| Simsiam + RSA | **63.7±0.5** | **89.2±0.5** | **84.0** |

## 5   Conclusion

In this paper, we first empirically demonstrate that two positive instances generated by the aggressive augmentations can cause the semantic shift issue, which introduces noisy pairs and degrades the quality of learned representation. To alleviate this issue, we propose a novel method RSA, which dynamically adjusts the weights of clean and noisy pairs based on the memorization effects. Experimental results show that our proposed method achieves state-of-the-art results on various datasets and consistently improves the generalization on a series of downstream tasks.

The main limitation of this paper is that, although we know there is noise from aggressive augmentations, the specific conditions under which noise occurs and how much DNNs will suffer from it are still unknown. We leave it as the future research direction.

## Acknowledgments and Disclosure of Funding

The authors would like to thank the anonymous reviewers and the meta-reviewer for their constructive feedback and encouraging comments on this work. Yingbin Bai was supported by CSIRO Data61. Erkun Yang was supported in part by the National Natural Science Foundation of China under Grant 62202365, Guangdong Basic and Applied Basic Research Foundation (2021A1515110026), and Natural Science Basic Research Program of Shaanxi (Program No.2022JQ-608). Zhaoqing Wang was supported by OPPO Research Institute. Bo Han was supported by the RGC Early Career Scheme No. 22200720, NSFC Young Scientists Fund No. 62006202, and Guangdong Basic and Applied Basic Research Foundation No. 2022A1515011652. Cheng Deng was supported in part by the National Natural Science Foundation of China under Grant 62132016, Grant 62171343, and Grant 62071361, in part by Key Research and Development Program of Shaanxi under Grant 2021ZDLGY01-03, and in part by the Fundamental Research Funds for the Central Universities ZDRC2102. Tongliang Liu was partially supported by Australian Research Council Projects DP180103424, DE-190101473, IC-190100031, DP-220102121, and FT-220100318.

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
