# OpenReview forum: "RSA: Reducing Semantic Shift from Aggressive Augmentations for Self-supervised Learning"
_NeurIPS.cc/2022/Conference — NeurIPS 2022 Accept_

### Official Review · Reviewer_mkyA · 2022-07-04

**Rating:** 4
**Confidence:** 4
**Soundness:** 3 good
**Presentation:** 3 good
**Contribution:** 2 fair

**Summary:**

This paper proposes a new learning strategy that assigns different weights for aggressively augmented data pairs at different training stages, to deal with the semantic shift problem caused by aggressive data augmentation methods for SSL.

**Questions:**

Some works do not have momentum encoder (e.g., SimSiam). Is that possible to generalize the proposed learning strategy to them ?

**Limitations:**

The novelty is limited, and some comparisons are missing. Please refer to the comments in the Weaknesses for more details.

**Strengths And Weaknesses:**

--Strengths：
1. This work contains extensive experiments to verify the effectiveness of the proposed strategy.

--Weaknesses：

1. Although this work has a good motivation, the proposed method is quite incrmental, which does not meet the standard of NeurIPS. For example, this work claims that  aggressive augmentations are retained, which is different from ReSSL [a] (lines 39-42). However, ReSSL does not give up the aggressive augmentations. ReSSL also feeds the weakly augmented images to the teacher model ( similar to the target network) , and inputs the aggressively augmented images to the student model (similar to the online network), which is exactly the same as MSR. The proposed learning strategy is more like a trick, which cannot support a paper to be accepted by NeurIPS.

2. According to Figure 3, the proposed strategy seems not to work very well on some datasets, as the increments brought by the decayed $\beta$ seems to be negligible.

3. As for the experiment, it would be great to show some results compared to ReSSL [a] and W-MSE 4 [b] with the same batch size and training epochs. Concerning the second weakness, it is not quite clear where the improvement comes from. Does it come  from the different training settings (The batch size usually has a large impact on the performance in SSL) ?


Overall, since I have some concerns about the effectiveness of the proposed strategy, the missing comparisons (I think it is hard to supplement them during the rebuttal stage), as well as the limited novelty, I vote for a rejection for this paper, but I may increase my score after the discussion stage, especially if I misunderstand something.


**Ref**

[a] Mingkai Zheng, Shan You, Fei Wang, Chen Qian, Changshui Zhang, Xiaogang Wang, and Chang Xu. Ressl: Relational self-supervised learning with weak augmentation. 2021.

[b] Aleksandr Ermolov, Aliaksandr Siarohin, Enver Sangineto, and Nicu Sebe. Whitening for self-supervised representation learning. In ICML, pages 3015–3024, 2021.

---

> ### Author Response · Authors · 2022-08-02
> **Response to Reviewer mkyA**
>
> Q1. ReSSL does not give up the aggressive augmentations. ReSSL also feeds the weakly augmented images to the teacher model … which is exactly the same as MSR.
>
> A1. Although our method and ReSSL both employ the aggressive and weak augmentation strategy, both the motivation and implementation of ReSSL and our method are quite different. Our method uses weak augmentation against the semantic shift problem of two instances from the same example, which widely exists in contrastive and non-contrastive SSL methods. And, ReSSL uses weak augmentation to find better relations between different examples, which does not proactively solve the semantic shift problem.
>
> Moreover, ReSSL replaces two aggressive augmentations with one weak and one aggressive augmentation, which reduces the diversity of augmented instances, missing the comparison of two aggressive augmented instances. In our method, we keep the comparison of two aggressive augmented instances and add a comparison of aggressive and weak augmented instances. We also dynamically adjust weights for different kinds of pairs.
>
> $ \rule[0pt]{17.5cm}{0.05em}$
>
> Q2. According to Table 3, the proposed strategy seems not to work very well on some datasets, as the increments brought by the decayed β seems to be negligible.
>
> A2. In table 3, our method outperforms the baselines by a large margin, with 2.8% for 100 epochs and 1.6% for 200 epochs. From table 1, 2, and 3, we can easily observe that our method consistently improves the performance on all datasets. I think you mean Figure 3, which compares the performance with different β. It may be not obvious in Figure 3, and we will separate the figure into three figures with their coordinates to make improvement clear. However, a good value of β can improve the performance, with +0.6% (total increments +0.7%) for STL-10 and +0.8% (total increments +1.7%) for CIFAR-100, which accounts for 86% of increments for STL-10 and 44% of increments for CIFAR-100. Overall, β is an important hyper-parameter to adjust neutralizing effects for different datasets. The digital numbers in figure 3 are in the table.
>
> | β | 0 | 0.2  | 0.3 | 0.4 | 0.5 | 0.7 |
> | ------ | ------ | ------ | ------ | ------ | ------ | ------ |
> | CIFAR-100 | 70.9 |  70.9 | 71.3 | 71.0 | 71.0 | 70.5 |
> | STL-10    | 93.7 |  94.1 | 94.3 | 94.3 | 94.3 | 94.3 |
> | ImageNet-100 | 81 |  83.2 | 83 | 83.7 | 83.5 | 82.5 |
>
>
> $ \rule[0pt]{17.5cm}{0.05em}$
>
> Q3. As for the experiment, it would be great to show some results compared to ReSSL [a] and W-MSE 4 [b] with the same batch size and training epochs.
>
> A3. We agree that the number of examples used in the loss computation is important for SSL methods. There are two types of methods to deal with it. One is directly training with a large batch size, e.g., BYOL, SimCLR, and W-MSE 4 that use a batch size of 4096 for ImageNet-1K. The other one employs a large memory buffer and saves recent embeddings instead of large batch sizes, e.g., MoCo and ReSSL. Although ReSSL uses a batch size of 256, it employs a memory buffer of 130k, which is 65 times the batch size of the proposed method.
>
> For a fair comparison, we compare our method with W-MSE 4 and ReSSL with the fixed computation resources (same number of backpropagations) on ImageNet-1K. The results of W-MSE 4 and ReSSL are directly cited from their papers.
>
> |Method | Epochs | Backprop  | Top1 |
> | ------ | ------ | ------ | ------ |
> | ReSSL | 200ep |  $\times$ 1 | 69.9 |
> | W-MSE 4 | 100ep |  $\times$ 2 | 69.43 |
> | Ours | 100ep |  $\times$ 2 | 71.4 |
>
> |Method | Epochs | Backprop  | Top1 |
> | ------ | ------ | ------ | ------ |
> | W-MSE 4 | 400ep |  $\times$ 2 | 72.56 |
> | Ours | 200ep |  $\times$ 2 | 73.1 |
>
> From the above tables, we can see that for 200 backpropagations (epoch $\times$ Backprop), the proposed method is higher than ReSSL and W-MSE 4, with about +1.5% and +2% respectively. Although W-MSE 4 does not give the result of 200 epochs, the proposed method with 200 epochs (400 backpropagations) is continually higher than W-MSE 4 with 400 epochs (800 backpropagations), with about +0.5% increments.
>
> $ \rule[0pt]{17.5cm}{0.05em}$
>
> Q4. Some works do not have momentum encoder (e.g., SimSiam). Is that possible to generalize the proposed learning strategy to them?
>
> A4. The proposed method does not rely on the momentum encoder. We have given the results with SimSiam in the supplementary.

---

> > ### Comment · Reviewer_mkyA · 2022-08-05
> > **Response to Rebuttal**
> >
> > Thanks for your response to my questions, but my questions are only partially addressed. So, currently, I still maintain my score.
> >
> > Q1. We all consider that the paper has a good motivation, but, tbh, the modifications proposed in this paper are quite incremental (Reviewer Ckh1  also mentions this). Such minor change is usually regarded as a small contribution in most papers published at top conferences. Using such minor change as the main contribution or novelty makes the whole paper look like a techniqual report, which could be more suitable for a workshop. From the novelty perspective, I do not think this paper can be accepted, especially concerning the high volume of submissions to NeurIPS this year.
> >
> >
> > Q2. Yes, it should be Figure 3. In fact, I do not care about the total increments over BYOL etc., since the training and backbone settings are different from those in the original papers.  But the experiments on different $\beta$ values are quite important. Accoding to the equation (5), $\beta$ value determines the weight of learning on aggressive augmented pairs during the whole training process, which is closely related to the motivation (lines 47-51).  But from Figure 3, the changes are quite small. In addition, large $\beta$ (e.g., $\beta$=0.7) means that the network first puts more weights on aggressively augmented data pairs and then the weights are reduced as the training goes, which is exactly the proposed strategy. But why is the performance of $\beta$=0.7 even lower ? I think these observations demonstrate that the proposed strategy does not solve the semantic shift problem well. Plus, is there any way to adjust  $\beta$ dynamically ? It seems to be difficult to try different $\beta$ values when very large datasets are used for training.
> >
> >
> > Q3. I still think that it is important to do experiments with the same settings (such as the same batch size). Moreover, I disagree with the comment "*Although ReSSL uses a batch size of 256, it employs a memory buffer of 130k, which is 65 times the batch size of the proposed method*". I think it is incorrect to regard the buffer size as the batch size. For example, SimSiam does not have a memory buffer, but MoCo-V1 [a] has. SimSiam still outperforms MoCo-V1 by a large margin with the same batch size.
> >
> >
> > I have to say that authors find a good perspective, namely solving the semantic shift problem, to improve the performance. But concerning the lack of some comparisions, the limited novelty, the above issues related to the proposed strategy, I think the paper does not achieve the standard of NeurIPS. I encourage authors to delve into this direction deeper, and I believe a good research paper usually needs more time.
> >
> > Ref:
> >
> > [a] He, Kaiming, et al. "Momentum contrast for unsupervised visual representation learning." Proceedings of the IEEE/CVF conference on computer vision and pattern recognition. 2020.

---

> > > ### Author Response · Authors · 2022-08-07
> > > **Response to Reviewer mkyA**
> > >
> > > Q1. the modifications proposed in this paper are quite incremental.
> > >
> > > A1. Although our method is simple, we do think it is an advantage, which means it does not require large modifications to the current SSL methods. In terms of novelty, we first introduce the memorization effects of DNNs to address this important problem. From the remarkable improvements, we can say the proposed method well addresses the semantic shift problem for SSL methods.
> > >
> > > $ \rule[0pt]{8cm}{0.05em}$
> > >
> > > Q2 (a). from Figure 3, the changes are quite small.
> > >
> > > A2 (a). Thanks, We would like to clarify that the visually small changes in Figure 3 are mainly due to the improper scale setting of y-axis.
> > >
> > > Actually, a good value of $\beta$ improves the results by +0.6% on CIFAR-100, +0.8% on STL, and  +2.7% on ImageNet-100.  Compared with the baseline performance, these improvements are all non-trivial.
> > >
> > >
> > > Q2 (b). why is the performance of \beta=0.7 even lower?
> > >
> > > A2 (b). $\beta$ is used to to re-weight the losses from weak and aggressive augmented pairs, and its value depends on the level of semantic shift. If there is no semantic shift, $\beta$ should be large and we make full of aggressive augmentations. If there is a large amount of semantic shift, we should set $\beta$ to be small and reply more on weak augmentations.
> > >
> > >
> > > Q2 (c). It seems to be difficult to try different values when very large datasets are used for training.
> > >
> > > A2 (c). Since we just introduce only one hyperparameter in this paper with a recommended value, "0.4 for large datasets", using our method on large datasets will not increase many experiments.
> > >
> > > $ \rule[0pt]{8cm}{0.05em}$
> > >
> > > Q3. I still think that it is important to do experiments with the same settings (such as the same batch size).
> > >
> > > A3. We strongly agree that experiments with the same settings are important. The settings, including batch size and epochs, for all the methods on CIFAR-10/100, STL-10, Tiny ImageNet, and ImageNet-100 are the same. For ImageNet-1K, we follow the settings in existing papers, such as SimCLR and BYOL, and compare our method with the best results obtained by the baseline methods.  we also provide the results with the same batch size for SimSiam and BYOL as follows. And for ReSSL, since it utilizes memory buffer to greatly reduce the batch size, we only compare its best results as in table 4.
> > >
> > > | Method  | Epochs | Batch size | Top1 |
> > > | ------- | ------ | ------ | ------ |
> > > | SimSiam | 100ep  | 256    | 68.1   |
> > > | SimSiam | 100ep  | 2048   | 67.9   |
> > > | Ours    | 100ep  | 2048   | 71.4   |
> > >
> > > | Method | Epochs | Batch size | Top1 |
> > > | ------ | ------ | ------ | ------ |
> > > | BYOL   | 300ep  | 4096   | 72.5   |
> > > | BYOL   | 300ep  | 2048   | 72.4   |
> > > | Ours   | 200ep  | 2048   | 73.1   |
> > >
> > > We can see that our method can consistently outperform SimSiam and BYOL under the same batch size.

---

> > > > ### Comment · Reviewer_mkyA · 2022-08-07
> > > > **Response to Rebuttal**
> > > >
> > > > Thanks for your quick response. I think I may have a different understanding about $\beta$ after reading your answer to Q2 (b). Could u please confirm the following:
> > > >
> > > > a.  Does large $\beta$ mean that the model makes full of aggressive augmentations at early stage of training ?
> > > >
> > > > I am also confused about your answer in A2(b), as it seems to be inconsistent with your lines 47-51 (or lines 12-14).  According to lines 47-51, it should be "**If there is a semantic shift**, $\beta$ should be large at the beginning of training"

---

> > > > > ### Author Response · Authors · 2022-08-07
> > > > > **Response to Reviewer mkyA**
> > > > >
> > > > > Thanks for your careful discussion.
> > > > >
> > > > > From equation 5, we can know that if $\beta$ is 1, it will conduct full of aggressive augmentations. When $\beta$ is close to 1, the contribution of weak augmentation becomes very small. Since $\beta$ start value ($\beta_{base}$) in figure 3 will affect the weights for weak and aggressive augmented pairs during the whole training process, it is important to set it properly to reduce the semantic shift problem.
> > > > >
> > > > > In the old version line 47-51, we claim "set a relatively large weight...the beginning of training". According to the memorization effects, the noisy impacts will be small at the beginning, then increase during the training. We decrease the weights of aggressive-aggressive pairs against increasing noisy impacts. So, the weights are large at the beginning and small at the end. Here, "a relatively large weight" is compared with the weights at the end, which is not related to the $\beta$ start value. For example, $\beta$ starts from 0.1, but 0.1 is still larger than 0 at the end. We will make this clear in the final version.
> > > > >
> > > > > Do you have any other concerns? If not, could you re-consider your recommendation? Thanks.

---

> > > > > > ### Comment · Reviewer_mkyA · 2022-08-07
> > > > > > **Response to Authors**
> > > > > >
> > > > > > I am grateful to authors' careful explanations, and now I understand the meaning of $\beta$ for the training. It would be great if authors can provide more details and explanations about $\beta$ in the new version to avoid misunderstandings.
> > > > > >
> > > > > > I slightly increase my score, due to the authors' helpful explantations about $\beta$ and my misunderstanding. But I still lean to rejection, since the issues related to novelty and the comparision with the ReSSL with the same training settings are not addressed. The novelty is weak and the comparision is also needed. I do not think using the memory buffer is an excuse to avoid the fair comparision with such strong and related baseline (see the example about MOCO-v1 and SimSiam in my previous response).

---

> > > > > > > ### Author Response · Authors · 2022-08-07
> > > > > > > **Thank Reviewer mkyA. However, we respectively do not agree with you on novelty**
> > > > > > >
> > > > > > > Dear Reviewer mkyA,
> > > > > > >
> > > > > > > Thanks very much for your prompt reply.
> > > > > > >
> > > > > > > We respectively do not agree with you on that the paper lacks novelty. Specifically, we cannot agree on your comment "minor change is usually regarded as a small contribution".  Simplicity is not equal to being incremental or not novel. A simple and effective algorithm will be appreciated and will interest many readers because it generalizes better.
> > > > > > >
> > > > > > > Regarding the comparison with SSL with the same setting, we think it is a minor issue because our method outperforms the well-tuned state-of-the-art methods. The effectiveness has already been shown. Given that we have also successfully reduced the side-effect of the semantic shift problem for data augmentation, the paper would interest many readers and benefit many partitioners.
> > > > > > >
> > > > > > > Do you reconsider your recommendation?
> > > > > > >
> > > > > > > Best wishes,

---

> > > > > > > > ### Comment · Reviewer_mkyA · 2022-08-07
> > > > > > > > **Last Response**
> > > > > > > >
> > > > > > > > No, I will not change my attitude about this paper, and I believe it is not good enough to be accepted. As for the novelty, I do not think such minor change can support a paper to be accepted by NeurIPS.  "simple and effective" does not mean "superficial  and boring idea". I cannot convince myself to accept a paper whose main contribution looks like a trick. Plus, I am not sure whether it is "effective" enough, as authors do not show fairly compare with the SOTA ReSSL (the improvement over BYOL is also not significant).

---

> > > > > > > > > ### Author Response · Authors · 2022-08-07
> > > > > > > > > **Response to the comparison with ReSSL**
> > > > > > > > >
> > > > > > > > > Thank you so much for your prompt reply.
> > > > > > > > >
> > > > > > > > > As you suggested, we compare the proposed method with ReSSL in the same setting, with a batch size of 256, and 200 epochs, and the same network architecture.
> > > > > > > > >
> > > > > > > > > |  Method | CIFAR-10| CIFAR-100 | STL-10 | Tiny ImageNet | ImageNet-1K |
> > > > > > > > > | ------- | ------- | ------ | ------ | ------ | ------ |
> > > > > > > > > | ReSSL   | 90.2    |  63.79 | 88.25  | 46.60  | 69.9   |
> > > > > > > > > | Ours    | 92.4    |  68.9  | 93.3   | 54.9   | 70.5   |
> > > > > > > > >
> > > > > > > > > Note that, we have not carefully turned the proposed method with a small batch size of 256 on ImageNet-1K, so we believe that with careful turning, the result will be better.
> > > > > > > > >
> > > > > > > > > Do you have any further suggestions?

---

### Official Review · Reviewer_Ckh1 · 2022-07-11

**Rating:** 5
**Confidence:** 3
**Soundness:** 2 fair
**Presentation:** 2 fair
**Contribution:** 2 fair

**Summary:**

The authors find that in most self-supervised learning methods, when applying aggressive augmentations to further improve the diversity of training pairs, there would exist severe semantic shift problem, thus degrading the transfer performance.

To address this	problem,  the authors propose a new SSL paradigm, which counteracts the impact of semantic shift by balancing the role of weak and aggressively augmented pairs.

With the training going on, the authors  gradually reduce the weights of aggressively augmented pairs.

Experiments have been done on the small datasets (CIFAR10/100) and medium datasets (STL-10  and Tiny ImageNet), and large datasets ( ImageNet-100 and ImageNet-1K), which have validated the effectiveness of the proposed method.

**Questions:**

see weakness.

**Limitations:**

yes

**Strengths And Weaknesses:**

The motivation of the proposed method is strong and clear. The authors fully consider the semantic shift problem, thus propose to minimize the negative impacts of noisy positive pairs from aggressive augmentations while taking advantage of aggressive augmentations. It has been achieved by different weights while training.

But the weakness is that, I don't think the novelty is strong enough. As for compared with the previous work BYOL, I can find that the only difference is to add an extra aggr. aug. stream. The main architecture almost keeps the same.

And for the re-weighting strategy, the authors use equ.(6) to achieve it. It is not flexible enough, which doesn't consider the quality of each augmentation pair.

And in Figure 2, is there a mistake? I think in the row three, v_m and v_a should be exchanged.

---

> ### Author Response · Authors · 2022-08-02
> **Response to Reviewer Ckh1**
>
> Q1. I don't think the novelty is strong enough ... the only difference is to add an extra aggr. aug. stream. The main architecture almost keeps the same.
>
> A1. In this paper, our target is not to design a new architecture of SSL methods but address the critical semantic shift problem caused by aggressive augmentation. To the best of our knowledge, this is the first work that uses the memorization effects of DNNs to address the semantic shift problem in SSL and achieve good results. We do think the proposed method will attract certain interests in the SSL community. Moreover, We do not think the simplicity of the algorithm implementation will decrease the novelty of our method. Instead, we consider it as one of our advantages.
>
> $ \rule[0pt]{17.5cm}{0.05em}$
>
> Q2. And for the re-weighting strategy, the authors use equ.(6) to achieve it. It is not flexible enough, which doesn't consider the quality of each augmentation pair.
>
> A2. Considering the quality of each augmented pair during training is a good idea. However, due to the diversity of training images and randomness of augmentation process, it is difficult to accurately measure the quality of different augmented pairs. Fortunately, DNNs have a memorization effect that DNNs tend to first fit clean examples (majority) and then overfit noisy examples (minority). Therefore, we propose to gradually reduce the weights for aggressive pairs, which will implicitly exploit the clean (high-quality) pairs and reduce the effect of noisy (low-quality) pairs.
>
> $ \rule[0pt]{17.5cm}{0.05em}$
>
> Q3. And in Figure 2, is there a mistake? I think in the row three, v_m and v_a should be exchanged.
>
> A3. Thanks for pointing out the mistake. We will fix it in the next version.

---

> ### Author Response · Authors · 2022-08-07
> **Response to Reviewer Ckh1**
>
> We believe that we have answered all the concerns. Do you need further clarifications? Thanks.

---

> ### Author Response · Authors · 2022-08-07
> **Rolling discussion**
>
> Dear Reviewer Ckh1,
>
> The rolling discussion period will be closed soon. Can you inform us if you have any remaining concerns? Thanks very much.
>
> Best wishes,

---

> ### Comment · Reviewer_Ckh1 · 2022-08-09
> **Response to Authors**
>
> Combining with the authors' rebuttal and other reviewers' comment,
> I have raised my score.

---

### Official Review · Reviewer_ofeU · 2022-07-11

**Rating:** 7
**Confidence:** 4
**Soundness:** 3 good
**Presentation:** 3 good
**Contribution:** 3 good

**Summary:**

The data augmentation transformations used in some self-supervised learning models for vision can generate pairs that are not semantically consistent. For example, cropping or blurring too aggressively can produce an image that is not identifiable as its class label. This work addresses this shortcoming by decreasing the degree to which a model relies on aggressive augmentations later on in training. The authors show doing so improves Top-1 linear classification accuracy for ImageNet as well as on object detection/segmentation for COCO over other self-supervised baselines.

**Questions:**

What's the intuition for the claim that "small datasets have higher probabilities of generating noisy samples”? I don’t see the connection with the size of the dataset and how augmentations act on those samples. Can you make this explicit?

**Limitations:**

Yes

**Strengths And Weaknesses:**

The authors highlight a problem arising when data augmentations are too aggressive—thus producing samples that not semantically meaningful ("noisy samples"). The authors propose a solution to address this problem by reducing the weight given for aggressive augmentations later on in training. The authors motivate doing so by citing recent work illustrating that deep neural networks overfit to noisier samples later on in training.

The authors define “weak” augmentation to be random crop and horizontal flip—while “aggressive” augmentations additionally include other color based augmentations such as blurring and color jitter. This delineation is 1) at odds with the original motivation since even “weak” augmentations can produce crops that are not semantically meaningful. Thus violating the motivation that only “aggressive” augmentations produce noisy samples,  2) at odds with more recent SSL methods (Masked Autoencoders, MAE) that only use “weak” augmentations. If we instead want “aggressive” to capture the extent to which the augmented sample is noisy, then a more precise approach is to control the magnitude of the cropping, degree to which gaussian noise is added, etc. Such a definition of an augmentation’s “aggressive” extent would more directly validate the authors’ claims.

I find the method an unnecessarily complicated (and memory-intensive) approach to achieve the stated goal of controlling the extent to which a model relies on aggressively augmented samples. Rather than introduce two asymmetric networks with twice the memory footprint, did the authors attempt to adjust the extend to which the samples fed into the original models were “aggressively” or “weakly” augmented? Or other simpler variants before settling on this two asymmetric network approach? At the very least, I'd like to see stronger motivation for the proposed method's implementation.

Overall the paper is well-organized and clearly written. I found the sentences motivating the impact of “noisy samples” on the online network around line 144 confusing. I think the wording here can be improved. Algorithm 1 would be easier to follow if it was self-contained—it doesn’t include the loss. The portion of the diagram in Figure 2 on the right with numerous arrows and boxes is not easy to follow.

Overall I find the experiments well-motivated and convincing (putting aside the "weak" versus "strong" augmentation definition). Appropriate baselines are used, authors assess sensitivity to beta, and show performance across several tasks. However, several of the experimental results (Tables 3, 4, 5) are used as evidence of MSR’s superior performance, but do not include error bounds.

The authors conclude MSR is a novel method to “property utilize aggressive augmentations.” If the motivation is that “aggressive augmentations” produce semantically noise samples, then the method’s success is instead to decrease models’ reliance on aggressive augmentations. The claim that is “property utilizes” or “neutarlizes the semantic shift problem” strikes me as an exaggeration. I suggest the authors appropriately qualify these claims.

The paper’s title suggests the proposed method directly improves models’ robustness. Instead, the approach balances the extent to which models rely on “aggressively” augmented samples during training. There are no experiments directly measuring robustness to augmentations. I suggest the authors amend the title to more directly reflect the contribution.

Overall, I find the problem of ensuring data augmentation produces semantically meanginful pairs for self-supervised learning important. The proposed method attempts to avoid overfitting to noisy samples shows some performance gains over existing baselines. On the other hand, I find the “aggressive” versus “weak” augmentation definition used here to miss the mark. I also find the proposed method to be needlessly complex based on the stated objective.

Minor:
- inconsistent notation: v and v' describe two augmented samples but their representations are designated using subscript z1 and z2. It's easier for readers to follow the same designation either subscript or prime is used for pairs throughout.
- line 96 typo: “by utilizes”
- line 157 typo: “To further against”

---

> ### Author Response · Authors · 2022-08-02
> **Response to Reviewer ofeU**
>
> Q1. “weak” augmentations can produce crops that are not semantically meaningful.
>
> A1. Thank you for these valuable comments. We agree that in some cases, weak augmentation may produce noisy samples, e.g., crops without objects, and there are some existing works addressing this issue by exploiting results from object detection approaches. In this work, we focus on reducing the semantic shift problem from other types of augmentations. Although weak augmentation can produce noisy pairs, augmentations, such as gaussian blur and color jitter, are more likely to produce semantic shifts. So, we call these types of augmentation "aggressive augmentation". We will add more explanations about their definition.
>
> $ \rule[0pt]{17.5cm}{0.05em}$
>
> Q2. I find the method an unnecessarily complicated … “aggressively” or “weakly” augmented?
>
> A2. The asymmetric architecture, including an online network and a target network, has been used in many SSL methods e.g., MoCo, and BYOL. In fact, the success of MSR is not relying on asymmetric architecture or the momentum branch. In the supplementary, we show MSR also works well with SimSiam, which uses a network with shared weights.
>
> In terms of twice memory footprint, many SOTA methods, such as SimCLR, SimSiam, BYOL, and Barlow Twins, forward each view of input images twice and optimized via a symmetrized loss to reduce the disk pressure. For a fair comparison, we follow this common setting with the same computation amount as previous methods.
>
> The motivation of the proposed method is to reduce noisy effects while keeping the benefits from aggressive augmentations. However, how to accurately select noisy pairs or define what extent is difficult. For example, some extent of augmentation may produce good instances for most of the samples in datasets but may produce noisy samples for some samples. Before adopting MSR, we have tried to simply reduce the extent of aggressive augmentation or remove some of the aggressive augmented examples but got bad results because most of the aggressive augmented pairs are clean and helpful.
>
> $ \rule[0pt]{17.5cm}{0.05em}$
>
> Q3. Several of the experimental results (Tables 3, 4, 5) are used as evidence of MSR’s superior performance but do not include error bounds.
>
> A3. A full training time of the proposed method, including linear evaluation, on ImageNet-1K is close to 30 GPU days, which is too computation expensive for us to provide “error bounds”. To verify the stabilization of the proposed method, we provide the deviation for small and medium datasets. Most self-supervised learning papers do not provide “error bounds” due to the high computation cost.
>
> $ \rule[0pt]{17.5cm}{0.05em}$
>
> Q4. The claim that “property utilizes” or “neutarlizes the semantic shift problem” strikes me as an exaggeration.
>
> A4. Thanks for this valuable comment. (1) We aim to properly utilize aggressive augmentation. Since aggressive augmentations can significantly enhance the sample diversity for positive pairs, exploiting it during training can greatly improve the model performance. However, as revealed in the paper, aggressive augmentations will also introduce the semantic shift problem for some examples as a side effect. Therefore we design our method to take advantage of the aggressive augmentation and reduce the side effect simultaneously. (2)To deal with the semantic shift problem, we utilize the memorization effect of DNNs:  DNNs tend first to fit clean examples and then overfit to semantic shifted examples. Therefore, we propose to dynamically reduce the weights of aggressive pairs to reduce (neutralize) the semantic shift problem.
>
> $ \rule[0pt]{17.5cm}{0.05em}$
>
> Q5. I suggest the authors amend the title to more directly reflect the contribution.
>
> A5. Thanks for your valuable suggestion. We will change the title to “RSA: Reducing Semantic shift from Aggressive Augmentations for Self-supervised learning”
>
> $ \rule[0pt]{17.5cm}{0.05em}$
>
> Q6. What's the intuition for the claim that "small datasets have higher probabilities of generating noisy samples”?
>
> A6. We are sorry for using the confusing word "small dataset". "small dataset" means the dataset with small size images, not the number of samples in datasets. If we use the same augmentation strategy for datasets with small size images and datasets with large size images, e.g., the same extent of resizing and blur, images from datasets with small size images are more likely to become too dim to be classified. In figure 1, the noisy samples from CIFAR-100 show much dimmer than that from ImageNet-1K. Furthermore, in our experiments, we found the best value of β is 0.3 for datasets with small size images while β is 0.4 for datasets with medium and large size images, which means datasets with small size images require more ‘clean’ effects from aggressive-weak pairs against noisy effects from aggressive-aggressive pairs.

---

> ### Comment · Reviewer_ofeU · 2022-08-05
> **Revisions not found**
>
> The authors do not seem to have updated the submission based on the responses. It would be helpful to have "explanations about the definitions," clarifications around "small datasets", the title and other points discussed incorporated in a revised draft.

---

> > ### Author Response · Authors · 2022-08-07
> > **Update revisions**
> >
> > Thanks for your valuable comments. We have revised all the issues and updated the manuscript accordingly.

---

### Official Review · Reviewer_gbda · 2022-07-12

**Rating:** 8
**Confidence:** 4
**Soundness:** 4 excellent
**Presentation:** 3 good
**Contribution:** 4 excellent

**Summary:**

This paper studied self-supervised learning. A semantic shift problem in the aggressive augmentations of self-supervised learning is considered. This paper inherits the memorization effect in tackling noisy labels, and gradually reduces the weights of aggressively augmented pairs. Extensive experiments verify the effectiveness of the proposed method.

**Questions:**

1.	From the descriptions in Line 72—79, in fact, some prior works have targeted the problem of noisy examples in aggressive augmentation. This paper is a complement of previous studies. Could the statement mean that the improvement by the proposed method comes from more general types of data augmentation? The intuition of why the proposed method is superior should be further explained.
2.	In Line 80-91, this paper states that it provides an indirect way to handle noisy samples, which is different from the methods in learning with noisy labels. Intuitively, when the noise rate is low, these methods can also work very well in coping with noisy samples. More descriptions of the advantages of this indirect way should be added.
3.	This paper claims that the proposed method would not bring too much computation cost. However, the descriptions in Line 127-132 are not clear to support the claim. More discussions should be added.
4.	From Eq. (5), weight decay performs on all pairs. Could we perform weight decay only on the semantic-shifted pairs?
5.	Expect for the classification task, it seems that the proposed method is not much superior in other tasks. More discussions are needed for this issue.
6.	The experiments on ImageNet-1K are more pivotal to verify the effectiveness of the proposed method. The ablation study should include the experiments on ImageNet-1K.

7.	Minor comments:
•	The definition of aggressive augmentations is somewhat confusing to me when it firstly occurs in “Abstract”. More explanations can be added.
•	In Related Work, the authors mentioned self-supervised learning learn representations from large-scale unlabeled data via pseudo labels, which is unfamiliar to me. The main reason is that the definition of pseudo labels commonly occurs in semi-supervised learning. Self-supervised learning is more related to unsupervised learning.
•	In learning with noisy labels, “noisy” means the combination of “clean” and “incorrect”. Thereby, the authors should use “noisy” to indicate “incorrect”.
•	Figure 2 could be revised to highlight the contribution of this paper.
•	The full names of “AA” and “AW” can be provided in the caption of Table 1.
•	Typos:
-	Line 68 He et al -> He et al.
-	Line 82 using first fit examples -> using first fitted examples
-	Line 96 utilizes -> utilizing
-	Line 115 severe semantic shift problem -> a severe semantic shift problem


**Limitations:**

See above weaknesses.

**Strengths And Weaknesses:**

Strengths:
1. Self-supervised learning is a practical and much important research topic in the community.
2. The motivation of this paper is clear. The semantic shift problem is common in multiple self-supervised learning methods and needs to be addressed.
3. Experimental results on benchmark datasets and real-world datasets show the effectiveness of the proposed method. Besides, ablation studies are provided to better understand the proposed method.


Weaknesses:
1. The writing and organization need to be improved to enhance this paper.
2. The intuition of the advantages of the proposed method should be supplemented.

The weaknesses of this paper are detailed below.

---

> ### Author Response · Authors · 2022-08-02
> **Response to Reviewer gbda**
>
> Q1. Could the statement mean that the improvement by the proposed method comes from more general types of data augmentation?
>
> A1. Prior works mainly focus on noisy examples caused by cropping augmentation, which may generate images with only background information. Since current object detection approaches can provide some position guidance for the object in images, prior works mainly aim to filter out the noisy examples caused by cropping augmentation. In this paper, we mainly focus on other augmentations, such as blurring and color jitter. Compare to cropping augmentation, It is much harder to directly filter out the noisy example for these augmentations. Our work alternatively tries to alleviate the effect of these noisy examples by exploiting the memorization effect of DNNs, which can be considered the main reason for the performance improvements.
>
> $ \rule[0pt]{17.5cm}{0.05em}$
>
> Q2. Intuitively, when the noise rate is low, these methods can also work very well in coping with noisy samples.
>
> A2. Thank you for pointing out this issue. We agree that current methods can deal with the noisy label learning problem with a low noise rate. In learning with noisy labels, samples with noisy labels are fixed, which are not altered by any data augmentation. In contrast, noisy pairs are randomly generated by aggressive augmentations over epochs. Because of this characteristic, the early stopping trick is difficult to mine noisy pairs in self-supervised learning. Besides, roughly removing noisy pairs is more likely to delete clean pairs by mistake, especially in the low noise rate case, which results in performance degradation. Based on the above analysis, some classical label noise methods (e.g., selecting confident examples) cannot handle well the semantic shift problem in self-supervised learning. We are sorry for the confusion caused by the improper statement in the original paper and will revise this issue carefully in the next version.
>
> $ \rule[0pt]{17.5cm}{0.05em}$
>
> Q3. The descriptions in Line 127-132 are not clear to support the claim.
>
> A3. Statements in line 127-132 elaborate on the computation issues by directly introducing clean pairs, and the solutions for these issues are provided in the following paragraphs in line 133-153.
>
> $ \rule[0pt]{17.5cm}{0.05em}$
>
> Q4. From Eq. (5), weight decay performs on all pairs. Could we perform weight decay only on the semantic-shifted pairs?
>
> A4. From Eq. 5, we can know as β decreases, the weights of aggressive-aggressive pairs decrease, but the weights of aggressive-weak pairs increase.
>
> We found that accurate noise extraction is very challenging. For example, some extends of augmentations will generate good pairs for most samples, but occasionally generate noisy pairs for some samples. Selecting many clean pairs incorrectly will lead to a performance decline. Fortunately, DNNs have a memorization effect that DNNs tend to first fit clean examples (majority) and then overfit noisy examples (minority). Therefore, we propose to gradually reduce the weights for aggressive-aggressive pairs, which will implicitly exploit the clean (high-quality) pairs and reduce the effect of noisy (low-quality) pairs.
>
> $ \rule[0pt]{17.5cm}{0.05em}$
>
> Q5. Expect for the classification task, it seems that the proposed method is not much superior in other tasks.
>
> A5. The proposed method also achieves good performance for object detection and instance segmentation tasks, especially for the 1 * schedule (a long-time refinement will decrease the advantages between pre-train models). Specifically, compared with the competitive method MoCo V2, our method improves the bb 50 and mk 50, with 1.6% and 2.7% respectively. To show advantages, we also compare the proposed method with baselines employing much more complex augmentation, e.g., SwAV and UOTA with multi-crop that several times more data than ours.  Although these methods experience much more data, the proposed method is also superior to them.
>
> $ \rule[0pt]{17.5cm}{0.05em}$
>
> Q6. The ablation study should include the experiments on ImageNet-1K.
>
> A6. We agree that more ablation studies on ImageNet-1K can better support our method. However, one full training with linear evaluation is close to 30 GPU days, which is too computation expensive for us to directly conduct many experiments on it. So, we first study our method and select hyperparameters on small and medium datasets and then conduct experiments on the large datasets. Another reason is that we found the best value of β for medium and large datasets, e.g., ImageNet-100, is consistent.

---

### Meta-Review · Area_Chair_FDBR · 2022-08-26

**Recommendation:** Accept
**Confidence:** Certain

**Metareview:**

This paper aims to improve SSL pretraining by adjusting the strength of augmentations applied at different points in training, providing a large number of aggressive augmentations early in training with this rate decreasing over time to prevent the model from overfitting to noisy examples. Using this approach, the authors demonstrate substantial improvements over prior methods. All reviewers recognized the soundness of the motivation and were generally convinced by the experiments, though there were some concerns about whether the approach is too incremental since it is relatively simple. I strongly agree with the authors that simplicity is not a downside of an approach, but rather a benefit, and the fact that the approach works with such a small modification makes it more likely that this result is not caused by an obscure mix of hyperparameters. I also note that the authors engaged extensively with the reviewers, providing a number of additional experiments comparing to other approaches and providing further tests of the impact of the hyperparameter they introduce. I think this is an worthwhile paper which will have impact going forward, and I recommend acceptance.

**Award:**

No

---

### Decision · Program_Chairs · 2022-09-14

Accept